# Characteristics of Compacted Fly Ash as a Transitional Soil

**DOI:** 10.3390/ma13061387

**Published:** 2020-03-19

**Authors:** Katarzyna Zabielska-Adamska

**Affiliations:** Faculty of Civil Engineering and Environmental Sciences, Bialystok University of Technology, 15-351 Bialystok, Poland; kadamska@pb.edu.pl

**Keywords:** California Bearing Ratio (CBR), compacted soil, fine content, fly ash, intergranular void ratio, mixed soil, moisture content at compaction

## Abstract

Cohesive and non-cohesive soils show a number of properties typical of a given category. Cohesive soils are characterized by cohesion, and the properties of compacted soils closely depend on moisture at compaction. However, many researchers have found the existence of so-called mixed or transitional soils. Compacted transitional soils, macroscopically recognized as non-cohesive, are characterized by mechanical properties and hydraulic conductivity which are strictly dependent on the moisture content at compaction. The aim of this work is to show the influence of the content of fine particles in fly ash on the variation of California Bearing Ratio (CBR) values as a parameter strictly dependent on initial compaction. The CBR values were interpreted in terms of moisture at compaction, void ratio and intergranular void ratio. Three different research samples were selected with fine contents of 45%, 55% and 70%; all samples corresponded in terms of grading with sandy silt. Fly ash containing only non-plastic fines behaved as cohesive soils despite the lack of plasticity. The CBR values decreased with increasing moisture at compaction or void ratio. The CBR values, plotted as a function of the intergranular void ratio, have lower penetration resistance together with fine content.

## 1. Introduction

The definitions of cohesive and non-cohesive soils are determined quite clearly according to the standards and current classifications. In the EN ISO 14688-1:2018 [1], cohesive soils are defined as fine soils on the basis of a macroscopic evaluation of the moist soil after demolding of the sample. If the sample retains its shape, the soil is assigned to fine-grained soils. Further macroscopic tests, such as plasticity, dilacity, silky touch, disintegration in water or ratio of drying, determine the dominant fraction as silt or clay. Subsequent macroscopic studies are carried out in order to determine the type of soil. The classification of the already non-binding PN-86/B-02480 standard [2] was also based on a macroscopic evaluation—the wet cohesive soil had to show plasticity, and the dried cohesive soil had to maintain a solidity of lumps when subjected to a pressure of greater than 0.01 MPa. In doubtful cases, it was possible to rely on the plasticity index (PI), which in the case of cohesive soils was to be greater than 1%. The soil type assessment was based on detailed macroscopic studies or granulometric analysis. In many English-language publications, the behavior of sands is identified by the content of fines (*f*_C_)—interpreted in accordance with ASTM D653 [3] as soil content with a grain size of less than 0.075 mm. According to ASTM D2487 [4], cohesive soils are determined on the basis of sieve analysis—the soil is cohesive (fine grained) if at least 50% of the dried soil content passes through the 0.075 mm sieve. This condition, i.e., grain sizes of less than 0.075 mm, is met by both clay and silt. A more detailed division of cohesive soils into types is made based on the plasticity index.

Usually, the division of a given soil into a group of cohesive or non-cohesive soils is determined by a number of soil properties typical of a given class. Cohesive soils show cohesion, and their properties after compacting are strongly dependent on moisture content at compaction. However, many researchers have found the existence of so-called mixed or transitional soils. According to Mitchell and Soga [5], the presence of a fine fraction in a given non-cohesive soil significantly influences its strength. Non-plastic silty soils behave like pure sands. Soils macroscopically evaluated as non-cohesive and not demonstrating plasticity may however exhibit properties of cohesive soils, with changes occurring not only in strength characteristics but also in deformation and filtration, which was found in the example of fly ash [6,7,8]. The phenomenon of cohesion resistance in non-cohesive soils is the result of suction in the soil, which is often associated with capillary suction. The smaller the grain size, the greater the resistance of cohesion, which is also known as apparent cohesion. Saturation or complete drying causes loss of cohesion resistance. Compacted transitional soils, macroscopically recognized as non-cohesive, are characterized by mechanical properties and hydraulic conductivity that are strictly dependent on the initial moisture content. Like cohesive soils, they show a variety of properties depending on the moisture at compaction and whether it is greater or smaller than optimum water content.

The authors’ earlier work [6,7,8] has shown that compacted non-plastic fly ash should always be evaluated considering the moisture content at compaction, as in the case of cohesive soils. The aim of the work is to show how the variable content of fine particles in fly ash influences the diversification of its mechanical parameters, which will be shown in the example of the California Bearing Ratio (CBR) of ash as a parameter that is strictly dependent on the initial compaction [6,7]. The obtained results will be helpful in the assessment of fly ash for use in earth construction.

## 2. Background

Extensive research on the effect of the addition of non-plastic fines in sands was performed in the case of liquefaction. Polito and Martin [9] conducted studies on the liquefaction of 16 soils composed by mixing three primary soils categorized by median grain diameter (*D*_50_): two with a sand grain *D*_50_ of 0.43 mm and 0.18 mm, and a non-plastic silt with a *D*_50_ of 0.03 mm. Cyclic triaxial tests of newly created soils were conducted in which the silt content was between 4% and 75%. It was found that a silt content in the range of 25–45% in most soils ensures continuous sample structure, the highest compaction and resistance to liquefaction. Based on research on more than 300 natural soils, Cubrinovski and Ishihara [10] found that the limit of structure change in sandy soil is a content of fine fraction equal to 30%.

Thevanayagam and Martin [11] determined the optimum silt content as 20–30% while testing the cyclic strength of Ottawa sand with a *D*_50_ of 0.25 mm and non-plastic silt with a *D*_50_ of 0.01 mm, mixed in different weight ratios, where the silt content was 0–100%. It was found that when adding silt to the sand, the maximum (*e*_max_) and minimum (*e*_min_) void ratio values (volume of voids to the volume of solids) initially decrease with increasing silt content. Next, an increase in *e*_max_ and *e*_min_ was observed, which occur when the silt fraction becomes the dominant fraction in the mixture. Along with the increase in the number of silt particles, sand grains float inside the matrix, silt particles dominate, and sand grains may behave as reinforcement.

It is worth noting that prepared mixtures of double-fractioned soils form discontinuous grained soils, and the higher the difference between bimodal distributions, the higher the ratio between the mean diameters of both fractions [12,13]. It should therefore be stated that individual soil behavior may result from the content of fine particles and the discontinuity of the grain size.

In pure sands, the stress is transferred directly through the interactions between the grains, however, in clay soils, physical and chemical forces play a greater role. The consequence of these differences in the behavior of soils during tests of their compressibility is the level of stress at which the soil reaches the steady state line (SSL), which is higher in the case of pure sands. Sandy-silt (mixed) soils are characterized by an SSL line with a higher inclination, as shown by studies on uniform sand with rounded grains [14]. The higher the silt content in the mixture, the higher the SSL inclination that is observed.

Kwa and Airey [15] determined the effect of the addition of non-plastic feldspar particles (< 0.075 mm) on the properties of quartzite sand. The authors performed tests on samples of well-grained, fully saturated soils, compacted with the standard Proctor method (at optimum water content) and minimally compacted, to determine the effect of fine particles on the location of the critical state line (CSL) on the e-log p′. The authors obtained curves with a slightly higher inclination in relation to the CSL curve of pure sand, and only the curve with a 100% content of fine particles showed a greater inclination. It was determined that a content of fine particles in a mixture of 40–60% causes “transitional” soil behavior. The sand–silt mixture reaches its maximum density at about a 25% content of fine particles.

However, attention should be paid here to obtaining possibly different behaviors of sands with the addition of non-plastic or plastic fine particles. In one-dimensional and triaxial compression tests on artificially treated soils with different clay contents, Nocilla et al. [16] found that the boundary value is an 8% content of particles smaller than 0.002 mm. The transitional soil showed different compression behavior in both one-dimensional and triaxial conditions, compared to samples with 3.5%, 25% and 45% clay particle contents. According to Lipiński [17], the content of the fine fraction within a range of 17–36% differentiates the behavior of cohesive and non-cohesive soil subjected to compression testing under oedometer conditions.

As presented above, many researchers point to a change in soil behavior with a specified addition of fine particles. Santucci de Magistris et al. [18] and Carrier [19] observed, however, that oedometric compressibility and silty sand stiffness determined in the resonant column depends on the moisture content at compaction, as in the case of cohesive soils. The dependence of Young’s modulus on the moisture content of silty sand at compaction is described in [20]. The authors thus questioned the findings of other researchers that the mechanical properties of compacted sands are independent of their moisture content at compaction.

The author’s research [6,7,8] has shown that compacted fly ash, despite the macroscopic similarity to non-cohesive soils, should always be assessed taking into account the moisture content at compaction (at a given compaction effort) as with compacted cohesive soils; not only providing the value of the compaction index, because all specified mechanical parameters and hydraulic conductivity are strictly dependent on the moisture content of the fly ash at compaction.

## 3. Materials and Methods

Studies were carried out on various research samples of a fly ash and bottom ash mixture, henceforth referred to as fly ash because there was only a vestige of bottom ash in the mix. The waste came from the combustion of bituminous coal in Bialystok Thermal-Electric Power Station and was taken from different locations of the dry storage yard.

X-ray diffraction (XRD) patterns of the tested fly ash specified the basic mineralogical composition as: quartz SiO_2_, mullite 3Al_2_O_3_·2SiO_2_ and calcite CaCO_3_. Fly ash fine grains were non-plastic—a 3 mm thread cannot be rolled at any water content. For the 23 different tested samples, the specific gravity of solids in the fly ash (*G*_s_) was in the range of 2.05–2.29 (Figure 1a). Generally, the higher the G_s_ value, the thicker the grain size of the fly ash. The median grain diameter that corresponds to 50% passing by weight (*D*50) was ranged from 0.022 to 0.090 mm (Figure 1b). Standard deviation for all *G*_s_ results was 0.08, and for *D*_50_ was 0.017.

The tested fly ash was a soil where 40–80% was able to pass through a 0.075 mm sieve. Thus, most of the soil met the criteria specified for fine-grained soil according to ASTM D2487 [4]. For further research, three different samples were selected—I, II and III—with fine contents of 45–70%. The physical parameters for the averaged fly ash research samples: median grain diameter (*D*_50_), content of fines (*f*_C_), specific gravity (*G*_s_) and specific surface (*S*_s_) are shown in Table 1.

Compaction parameters (Table 2)—optimum water content (*w*_opt_) and maximum dry density (*ρ*_d max_)—were carried out according to the standard Proctor (SP) and modified Proctor (MP) methods. The particular research samples differed in specific gravity and grading, but all samples corresponded in terms of grading with sandy silt.

The main studies have been conducted as California Bearing Ratio (CBR) tests. CBR is expressed as the percentage ratio of unit load (*p*), which is applied so that a standardized circular piston may be pressed into a soil specimen to a definite depth with a rate of 1.25 mm/min and standard load, corresponding to the unit load (*p*_s_) necessary to press the piston at the same rate into the same depth of a standard compacted crushed rock:(1)$$CBR=\frac{p}{p_{s}}100{\%}_{\;}$$

The CBR value was used for the estimation of the subgrade or subbase strength and may be applied to evaluate the resistance to failure or indicate the load-carrying capacity. It should be noted here that CBR values in pavement design do not reflect the shear stresses that are generated due to repeated traffic loading. The shear stress depends on many factors; none of them are fully controlled or modelled in the CBR test [21]. Nevertheless, the California Bearing Ratio test has been widely used in soil and granular material testing in highway laboratories for over seventy years. The CBR method is still used as the basic method of pavement design in many countries or even as the recommended method for characterizing subgrades [21]. CBR values are closely connected with the characteristics of compaction, so the CBR test can be used as a method of earth work assessment [7].

The laboratory CBR tests were carried out to establish a relationship between the bearing ratio and fly ash compaction. Studies were performed on samples of fly ash compacted according to the standard Proctor (SP) and the modified Proctor (MP) methods of optimum water content *w*_opt_ ± 5%. The range of water content corresponds to the moisture of fly ash built into road embankments as well as into sealing barriers. The CBR tests were conducted on specimens directly after compaction and after their maximum swelling caused soaking (SAT) in water. Specimens were loaded with the ASTM D1883 [22] recommended load of 2.44 kPa (4.54 kg) during penetration tests. The greater CBR value was accepted as a result of calculation based on pressing piston resistance, represented in a given depth of 2.5 or 5.0 mm.

Additionally, maximum and minimum void ratio tests were conducted by applying vibration in order to evaluate the fly ash samples. The maximum void ratio (*e*_max_) was determined by pouring soil into the cylinder through a funnel, and the minimum void ratio (*e*_min_) by the compaction of the soil using vibration forks.

## 4. Results and Discussion

According to the literature [10,11], initially adding fine particles to the larger sand grains leads to a decrease in the volume of voids, since the smaller particles fill in the voids between the larger ones. After obtaining the optimum percentage of smaller particles, corresponding to the highest density of mixed soil, further addition of the fine results in a reverse trend. The volume of voids increases with the percentage of the smaller fraction. The larger grains are pushed away from each other and are gradually replaced by fines. The evaluation of transitional soils is carried out on the basis of the parameter e_s_, also known as the intergranular or skeleton void ratio (an index of active coarse-granular contacts). The distinction between void ratio and intergranular void ratio is to determine the inactive section of transitional soil—fines, which do not take part in the transmission of contact friction forces in the shearing process or have a secondary role. The simplified formula [23] has the form:(2)$$e_{s}=\frac{e+f_{C}}{1-f_{C}}$$
where *e*_s_ is the intergranular or skeleton void ratio, *e* is the void ratio, and *f*_C_ is the content of the fines presented in a decimal fraction.

Figure 2a presents the correlation between the intergranular void ratios calculated from the Formula (2) and fine particles content (in the case of fly ash—silt particles). The figure shows the lines of the maximum and minimum void ratios determined with the vibration fork method and the void ratios of soil samples compacted in CBR molds with the Proctor methods. Figure 2b shows the intergranular void ratio dependence on the void ratio obtained for each fly ash sample, differing in grain size and fine content.

The shape of the graph (Figure 2a) shows that the optimum fine particle content is exceeded in all tested fly ash specimens. The intergranular void ratio is many times larger than the void ratio (Figure 2b), and the difference increases with the rise in fine content. The high content of the fine fraction means that in this case, the intergranular void ratio will be larger than the maximum void ratio of coarse fly ash grains—greater than 0.075 mm, i.e., the silty particles dominate and the sand grains may act as matrix reinforcement.

Figure 3 shows the relationship between the CBR values and the moisture content at compaction, found in laboratory tests for sample I, which was 70% fines. The specimens tested without soaking reached the highest CBR values at *w* = *w*_opt_ − 5% moisture content, and the soaked ones at *w* ≤ *w*_opt_. The CBR values of fly ash tested after four days of soaking in water (after maximum swelling) depended on the initial moisture of the specimens (moisture content at compaction). However, they did not depend on the moisture content of the saturated samples in both compaction methods [6]. All relationships for both non-soaked and soaked specimens show that the CBR definitely depends on the moisture content at compaction, despite the lack of plastic particles in fly ash. Figure 3 presents scatter charts of CBR = *f*(*w*) with a quadratic polynomial regression fit. The data were subjected to a statistical analysis utilizing polynomial regression, where the R^2^ was in the range of 0.6201–0.9359.

Particular attention should be drawn to the visible effect of compacting energy on the CBR values of samples with the same initial moisture content yet compacted with different energy values. Specimens compacted with the modified Proctor method, with initial moisture content (*w* > *w*_opt_), were characterized by much smaller CBR values than specimens with the same moisture but compacted with the standard method (*w* < *w*_opt_). A similar relationship was obtained in the case of shear wave velocity in compacted silty sand [24].

A summary the CBR correlations based on all 140 CBR test results, obtained for I, II and III fly ash research samples differing in fine content, depending on the moisture content at compaction, void ratio and intergranular void ratio are shown in Figure 4. 

Figure 4a,b shows a visible effect of the tested fly ash moisture content and void ratio on the CBR value when different samples of fly ash compacted by both methods are compared. Generally, CBR decreases as compaction moisture content increases and dry density drops (void ratios increase—see Figure 4b). The CBR values are twice as big in sample I (*f*_C_ = 70%) compared to samples II (*f*_C_ = 45%) and III (*f*_C_ = 55%). The CBR value decrease is influenced by the tested fly ash graining and thus the optimum water content value (*w*_opt_) defined for each fly ash research sample. Sample II, featuring the finest graining (*f*_C_ = 45%) and reaching the lowest void ratio, produces the lowest CBR values. CBR = *f*(*w*) relationships are similar in shape; they all can be described by means of square trinomial curves reaching their maximum values at abscissa values similarly located relative to the optimum moisture content values for each of the samples.

Figure 4c shows the interpretation of the CBR values depending on the intergranular void ratio, taking into account intergranular contacts between grains thicker than 0.075 mm.

If the CBR values are plotted in terms of their moisture at compaction or void ratio, as shown in Figure 4a,b, the CBR values decrease with increasing compaction parameters. If the CBR values are plotted as a function of the intergranular void ratio, as shown in Figure 4c, the sand–silt mixtures give a lower penetration resistance along with fine content. The boundaries between research samples are very clear, and the influence of fine content on the structure is quite visible. The CBR values decrease in proportion to the growth of the fine fraction. Sand grains, in the case of sample I, act as reinforcement of the matrix. With a higher content of fine particles, this reinforcement disappears. The presented interpretation of CBR tests on fly ash samples characterized by different content of fines, depending on the intergranular void ratio, shows the importance of this parameter for the assessment of non-cohesive fine soil. In future, the author’s research will focus on other fly ash mechanical parameters and hydraulic conductivity, interpreted depending on the amount of fine content. Studies carried out on fly ash samples with different contents of fines will allow an indication of the fly ash graining at which the mechanical properties decrease rapidly, which can be important in the incorporation of ash in earthworks.

## 5. Conclusions

The author’s earlier work has shown that compacted fly ash, regardless of the macroscopic likeness to non-cohesive soils, should always be evaluated considering the moisture content at compaction. In order to explain this phenomenon, a study of three different research samples of varying fine content was shown. Tests were interpreted in terms of moisture content at compaction, void ratio and intergranular void ratio. The following conclusions regarding the effect of non-plastic fines on CBR test results were drawn:Grain-size distribution of the 23 different research samples of fly ash showed that 40–80% passed through the 0.075 mm sieve, so most of them meet the criteria specified for fine-grained soil.Three selected research samples, comprising 45%, 55% and 70% non-plastic fines, exceeded the optimal value of fine particles in the case of all samples, and behaved as cohesive soils despite the lack of plastic particles.The CBR values decrease proportionally with the increase of the fine fraction. Sand grains, in the case of the test with *f*_C_ = 45%, reinforce fly ash. This effect disappears with a higher content of fine particles.The CBR values, interpreted in terms of moisture at compaction or void ratio, decreased with increasing compaction parameters. The CBR values, plotted as a function of the intergranular void ratio, have a lower penetration resistance together with fine content.

## Figures and Tables

**Figure 1 materials-13-01387-f001:**
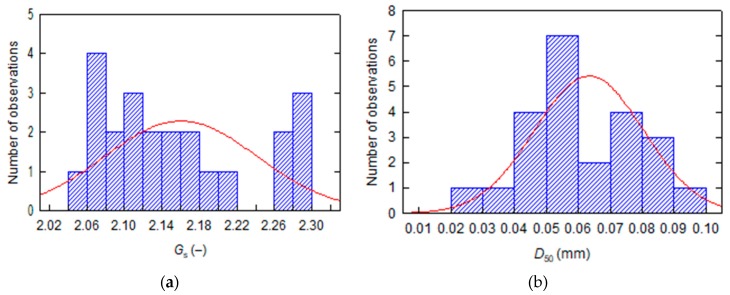
Distribution of fly ash parameters with fitted curves of normal distribution: (**a**) specific gravity of solids, (**b**) median grain diameter.

**Figure 2 materials-13-01387-f002:**
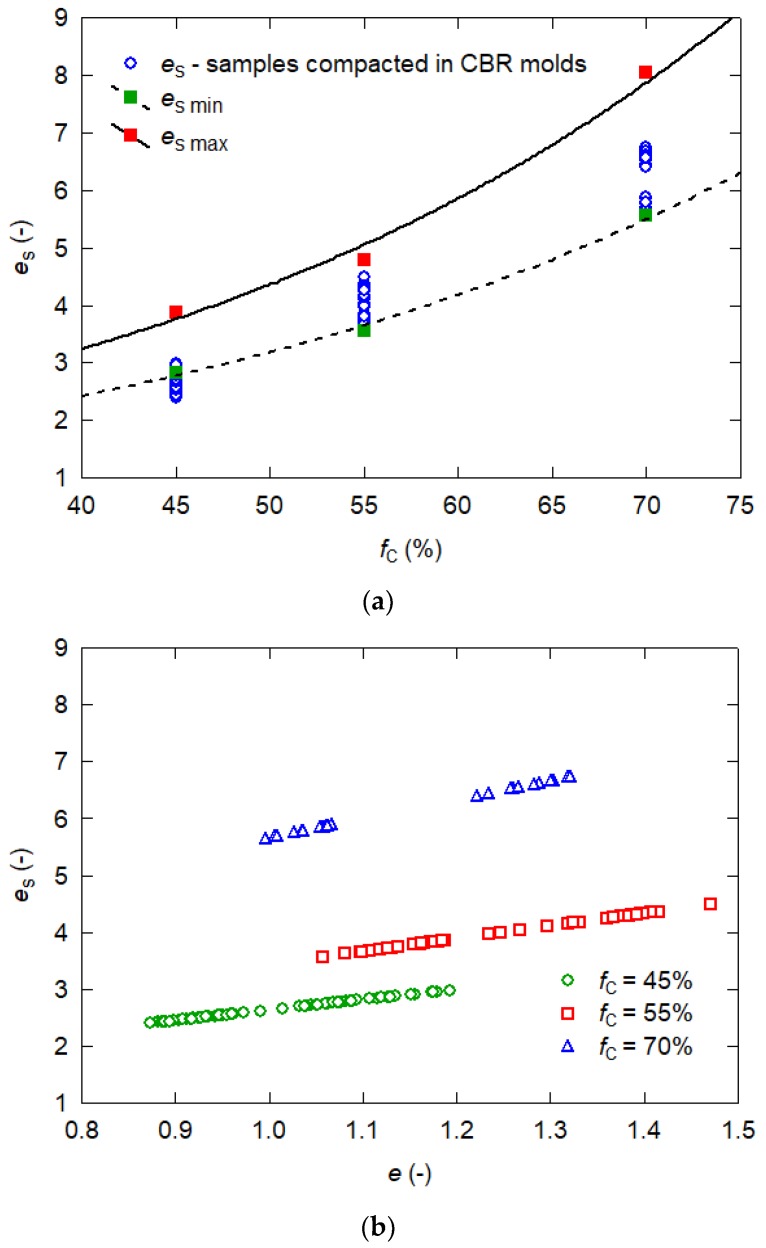
Intergranular void ratios versus (**a**) fine content in tested fly ash samples, (**b**) void ratios in tested fly ash samples.

**Figure 3 materials-13-01387-f003:**
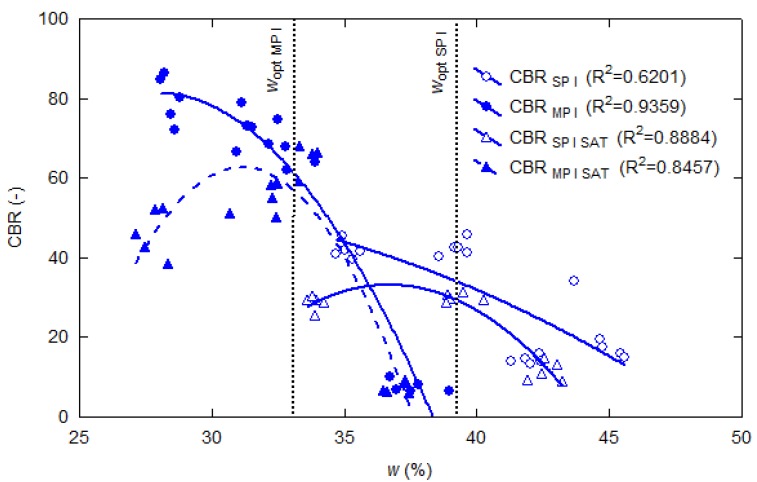
California Bearing Ratio (CBR) test results versus moisture content at compaction obtained for sample I, tested directly after compaction according to the standard (SP) and modified Proctor (MP) methods, and compacted and soaked (SAT) samples.

**Figure 4 materials-13-01387-f004:**
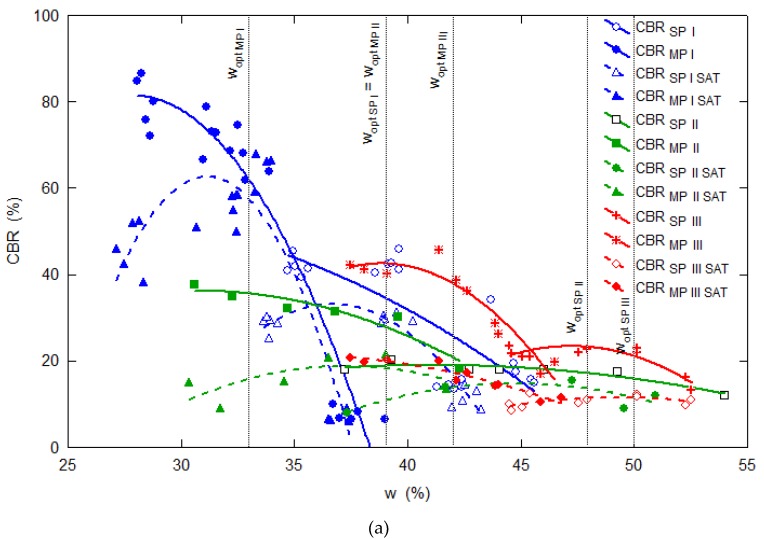
CBR test results obtained for three different fly ash research samples (I, II and III), tested directly after compaction by the standard (SP) and the modified Proctor (MP) methods at *w*_opt_ ± 5% or compacted and soaked (SAT), depending on (**a**) moisture content at compaction, (**b**) void ratio, (**c**) skeleton void ratio.

**Table 1 materials-13-01387-t001:** Physical parameters of averaged fly ash research samples.

Sample Number	*D*_50_ (mm)	*f*_C_ (%)	*G* _s_	*S_s_* (m^2^/g)
I	0.08	45	2.28	2.48
II	0.05	70	2.08	21.01
III	0.07	55	2.15	—

**Table 2 materials-13-01387-t002:** Compaction parameters of tested samples.

Sample Number	SP Compaction Method		MP Compaction Method	
	*w*_opt_ (%)	*ρ*_d max_ (Mg/m^3^)	*w*_opt_ (%)	*ρ*_d max_ (Mg/m^3^)
I	39.0	1.130	33.0	1.230
II	46.0	0.948	37.0	1.065
III	50.0	0.950	42.5	1.032

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
