# Peer review of "Characteristics of Compacted Fly Ash as a Transitional Soil"

_materials, 2020, doi:10.3390/ma13061387_

Round 1
Reviewer 1 Report
I do not question the legitimacy of the research carried out, I do not question the methodology used, but the presentation method resembles a technical report rather than developing of scientific study, but in my opinion this paper requires very significant corrections.
Comments:
The use of abbreviations in the text significantly impedes reading and understanding of the described relationships. A similar effect is caused by analyzing the graphic presentation of results with the accumulation of abbreviations that were not previously introduced, discussed, described.
There is no comments regarding the curves plotted on figures 3 and 4a.There is no explanation whether they are model curves or empirical curves or trend lines. In addition, why were not similar curves drawn on figures 4b, c?
It is necessary to unify the terms interchangeably used "grain" and "particle", I propose to decide on one formula,Placing charts in the "results" section without analyzing this graphical presentation of results is a bad idea.Reading the "discussion" section with no easy access to results is a significant impediment.
Consider changing the way the results are presented, as a complement to the results tables in paper or as an attachment.
Author Response
Dear Sir,
I would like to thank for all the remarks on my manuscript. The paper after reorganizing and revision is more comprehensible.
In reply to the individual comments:
According to the remark about using abbreviations in the text which impede reading and understanding the text, I have checked step by step the all abbreviation use. In all places the abbreviation had been preceded by a full name of the parameter or test method. For better explanation I have added definition for void ratio (lines 77-78) and definition and formula for CBR (lines 147-152).
I was stated stated in the review that there were no comments regarding the curves plotted in Figures 3 and 4a. He did not know if they were model curves or trend lines. Both figures present points being research results and fitted lines describing obtained relations. For better explanation I added statement “obtained in laboratory tests” in description of the Figure 3. In Figure 3 values of R2 were completed and I added some explanation about statistical analysis in lines 211-213. I expanded the description in Figure 3 caption about test methods. In Figure 4a description I added that figure presents results of laboratory test obtained for: I, II and III research sample. I emphasized that the parameter es is calculated (line 188).
Next remark was a question - why “were not similar curves on Figures 4b, c?”. Figures 4b and c show relationships between CBR and void ratio and skeleton (intergranular) void ratio, relatively. I had written in the paper (now lines 209-211) that “All relationships for both non-soaked and soaked specimens show that the CBR definitely depends on the moisture content at compaction, despite the lack of plastic particles in fly ash”. We do not have visible relationship between CBR and dry density (void ratio). In the Background (lines 117-121) I had explained that: “The author's work [6−8] has shown that compacted fly ash, despite the macroscopic similarity to non-cohesive soils, should always be assessed taking into account the moisture content at compaction (at a given compaction energy), similar to compacted cohesive soils, and not only providing the value of compaction index, because all specified mechanical parameters and hydraulic conductivity are strictly dependent on moisture at fly ash compaction process.”
I considered proposition to decide on one formula instead two: grain and particle. I tried to choose one on them, but in my opinion it is impossible. So, I have tried to set in order them grain is always connected with sands or grain-size distribution, particle ̶ with clay or fine particles. I looked at the literature on the subject, quoted in the paper, the use of these words is similar as in my paper.
According to review I decided to combine two chapters – the Results and Discussion. Earlier version was prepared according to journals’ recommendations. I did not change suggested change of result presentation on tables or appendix, because none of
other three Reviewers did not point out that. In my opinion presentation of test results in figures are the best for this type of paper.
Reviewer 2 Report
The aim of this paper is to study the characteristics of compacted fly ash as a traditional soil. I recommend to accepts the paper after minor revisions.
The comments have been included in the pdf.

Author Response
Dear Sir,
I would like to thank for all the remarks on my manuscript. The paper after reorganizing and revision is more comprehensible.
In reply to the individual comments:
The review was noted directly on the pdf copy of my paper. It was: “It is necessary to report the error of the equation”. It concerned to Figures 1, 2a and 3.
In Figure 3 values of R2 were completed and I added some explanation about statistical analysis in the text in lines 211-213.
In Figure 2 ̶ lines are not trend lines – they are given for shoving the border of maximum and minimum void ratio. Because only three samples were shown, each line was drawn on the basis of three points, so statistical analysis is impossible.
In the case of Figure 1, lines are fitted curves of the normal distribution, which was added to caption of Figure 1. Additionally, values of standard deviation were given for specific gravity of solids and median grain diameter results in lines 132-133.
Reviewer 3 Report
This is a of high quality manuscript and of very interesting subject. after minor revisions i suggest to be published.
line 58, 117: Please rephrase:the author's
line 142: it could be better if you could rephrase "by the" with "according to".
line 142: correspond to and not correspond in
I would suggest a small restructuring of the work. More specifically, I suggest that the results and discussion part to be together and that the diagrams to be close to the text so that it is easier to study the manuscript.
Author Response
Dear Sir,
I would like to thank for all the remarks on my manuscript. The paper after reorganizing and revision is more comprehensible.
In reply to the individual comments:
One of the remarks concerns rephrase “by the” with “according to” – it was done.
The second – replacing “the author’s work” – it was not done ̶ this statement does not contain an error, although it is perhaps not very elegant.
The third remark concerns replacing “correspond in” to “correspond to”. In text it is not possible (now lines 144-145), because it is “correspond in terms of...”.
According to next remark I decided to combine two chapters – the Results and Discussion. Earlier version was prepared according to journal’s recommendations.
Reviewer 4 Report
The paper presents the behavior of a soil depending on the content of fine particles in fly ash on the variation of CBR. It shows interesting results, but the aim of mixing fly ash in soils is necessary to be clearly exposed.
The first paragraphs in the introduction (lines 25-57) are a very good explanation about the classification of the soils, with regard to the usual division: cohesive and non-cohesive. It is one of the best I have read in an article. Congratulations. Then, in the second section, background, the author presents other researchers’ work in the field, with clear exposition of their advances.
Nevertheless, from my point of view, one idea is missing. It is necessary to indicate the aim of introducing fly ash in soils. Fly ash, as indicated in section Materials and Methods (it should be numbered as 3), is a waste material coming from thermal electric power stations. It should be mentioned that in flexible pavements, a bituminous layer is extended over a granular base and subbase, both over a compacted subgrade. However, due to the increasing traffic loads aggregates in bases and subbases (and even in the subgrade) must be treated with cement, with bitumen or another stabilizing agent, such as lime, fly ash or hydraulic road binder (Fedrigo 2019a; Linares-Unamunzaga et al. 2019; Wang and Baai 2020). In line 102 it is said that researchers study the behavior of obtaining different sands, but without including the aim of it. This idea is vital for the article because after all the introduction and the description of the materials, in line 146 it is the first time that is commented that the inclusion of fly ash is with the aim of improving some characteristics of the subgrade or subbase. It must be clearly presented in the introduction and summarized in the abstract.
Other minor points:
Line 152-154. CBR continues to be used as a basic method for pavement design in many countries. Any reference should be included.
Line 176. The parameter “es” must have the “s” as a subindex, as in Equation 1.
REFERENCES
Linares-Unamunzaga et al. (2019). Flexural Strength Prediction Models for Soil–Cement from Unconfined Compressive Strength at Seven Days. Materials, 12(3), 387.
Fedrigo ET al. (2019a). Flexural behaviour of lightly cement stabilised materials: South Africa and Brazil. Road Materials and Pavement Design, 10.1080/14680629.2019.1634637.
Wang and Baaj (2020). Treatment of weak subgrade materials with cement and hydraulic road binder (HRB). Road Materials and Pavement Design, doi: 10.1080/14680629.2020.1712224
Author Response
Dear Sir,
I would like to thank for all the remarks on my manuscript. The paper after reorganizing and revision is more comprehensible.
In reply to the individual comments:
The main suggestion cannot be used, because the content of the article was misunderstood. In review it was written about mixing fly ash in soils, which was not described in any part of my paper. I wrote about mixing different soils in the Background, where authors wanted to get soil with specific grain-size distribution, but fly ash that was a subject of my paper was used alone. I tested different samples of fly ash, which was a cause of various its granulation. Similarly, there are no any indications that fly ash is tested to be used as subgrade beneath of flexible pavements. My intention was not to narrow down the subject of the article. I stated in lines:
62-63: “The obtained results will be helpful in the assessment of fly ash for use in earth construction”.
164-165: “The range of water content corresponds to moisture of fly ash built into the road embankments as well as into sealing barriers”.
255-257: “Studies carried out on fly ash samples with different content of fines will allow an indication the fly ash grading at which the mechanical properties decrease rapidly, which can be important in the incorporation of ash in earthworks”.
Therefore, I do not see the need to expand the article with the indicated content and cite the given new articles. However, I would like to thank for an effort to improve my article.
The next remarks about adding literature item in lines 156-160 (now), improving the notation of parameter es (now line 181) and chapter numbering were used gratefully.
Round 2
Reviewer 1 Report
Dear Author,
I take note of your explanations for the review. The changes made to the manuscript are satisfactory and have certainly improved the quality of the paper.
I maintain my comments and suggestion change of result presentation on tables or appendix. The author, of course, has the right to make his own decision taking into account the suggestions (publisher's suggestions).
I do not present substantive comments
The final decision regarding publication is left to the Editors
Kind regards,
Author Response
Dear Sir,
I would like to thank for all the remarks on my manuscript. The paper after reorganising and revision is more comprehensible.
Yours faithfully,
Katarzyna Zabielska-Adamska
Reviewer 4 Report
It is true that in the section “Background” of the article it was commented about mixing different soils to get soils with specific grain-size. It is also true that it is not indicated that fly ash is tested for being used as subgrade in flexible pavements. Moreover, the aim of the paper is clearly stated in the abstract: “the aim of the work is to show the influence of the content of fine particles in fly ash on the variation of California Bearing Ratio (CBR). It is an interesting idea.
However, since the fly ash is a waste product, in my humble opinion, it is interesting and necessary to clearly indicate the application of the fly ash. It is also true that possible application are mentioned in the lines indicated (62-63: in earth construction; 164-165: road embankments, sealing barriers, 255-257, earthworks). I only suggest that it is better to underline more possible applications of the results of this article, even in the abstract. I just take some of the recent examples about treating base or subbase layers in pavements to show that other authors also research about including additives (cement, lime, etc.) in soils to increase their properties with a specific application. I think that highlighting the possible applications of the findings of this article would reach to a broader audience. Even more keywords could be added.
Author Response
Dear Sir,
I would like to thank for all the remarks on my manuscript. The paper after reorganizing and revision is more comprehensible.
I has added four new keywords and all keywords were given in alphabetical order.
Yours faithfully,
Katarzyna Zabielska-Adamska